# Identification of Structural and Molecular Signatures Mediating Adaptive Changes in the Mouse Kidney in Response to Pregnancy

**DOI:** 10.3390/ijms23116287

**Published:** 2022-06-03

**Authors:** Jorge Lopez-Tello, Maria Angeles Jimenez-Martinez, Esteban Salazar-Petres, Ritik Patel, Amy L. George, Richard G. Kay, Amanda N. Sferruzzi-Perri

**Affiliations:** 1Centre for Trophoblast Research, Department of Physiology, Development and Neuroscience, University of Cambridge, Cambridge CB2 3EG, UK; e.salazar.petres@gmail.com (E.S.-P.); rap67@cam.ac.uk (R.P.); 2Department of Animal Medicine and Surgery, Veterinary Faculty, Complutense University of Madrid, 28040 Madrid, Spain; mariadji@vet.ucm.es; 3Wellcome-MRC Institute of Metabolic Science, Addenbrooke’s Hospital, Cambridge CB2 0QQ, UK; a.l.george2@newcastle.ac.uk (A.L.G.); rgk27@medschl.cam.ac.uk (R.G.K.)

**Keywords:** pregnancy, kidney, physiology, signalling, renal medulla, mouse urine, mass spectrometry, keratins, proliferation, interstitial cells

## Abstract

Pregnancy is characterized by adaptations in the function of several maternal body systems that ensure the development of the fetus whilst maintaining health of the mother. The renal system is responsible for water and electrolyte balance, as well as waste removal. Thus, it is imperative that structural and functional changes occur in the kidney during pregnancy. However, our knowledge of the precise morphological and molecular mechanisms occurring in the kidney during pregnancy is still very limited. Here, we investigated the changes occurring in the mouse kidney during pregnancy by performing an integrated analysis involving histology, gene and protein expression assays, mass spectrometry profiling and bioinformatics. Data from non-pregnant and pregnant mice were used to identify critical signalling pathways mediating changes in the maternal kidneys. We observed an expansion of renal medulla due to proliferation and infiltration of interstitial cellular constituents, as well as alterations in the activity of key cellular signalling pathways (e.g., AKT, AMPK and MAPKs) and genes involved in cell growth/metabolism (e.g., *Cdc6, Foxm1* and *Rb1*) in the kidneys during pregnancy. We also generated plasma and urine proteomic profiles, identifying unique proteins in pregnancy. These proteins could be used to monitor and study potential mechanisms of renal adaptations during pregnancy and disease.

## 1. Introduction

Pregnancy is a physiological state accompanied by changes in multiple organs and systems of the mother’s body that enable correct growth of the foetus, in addition to guaranteeing maternal well-being [1,2]. Several adaptations occur in the kidney in response to gestation. For example, in response to maternal systemic vasodilation, there is an increase in the renal plasma flow and glomerular filtration rate (GFR) by 80% and 50%, respectively [1,3,4,5,6]. The renin–angiotensin–aldosterone system, which is central to blood pressure regulation, electrolyte–fluid homeostasis and acid/base handling, is also altered in pregnancy. In particular, there is a 5- to 10-fold rise in circulating renin concentrations linked to enhanced conversion of prorenin to renin by the maternal kidneys [7,8,9,10,11]. There are also structural changes in the maternal kidneys that are induced by pregnancy. In women, during pregnancy, kidney volume increases by approximately 30% [3], which is attributed to the increased vascularisation, interstitial volume, plasma flow and dilatation of the collecting systems, causing physiological hydronephrosis and hydroureters in approximately 80% of human pregnancies [3,12].

Defective adaptations in the maternal renal system can cause renal dysfunction and kidney disease during pregnancy, which can evolve into severe pregnancy complications for the mother, putting fetal and maternal health at severe risk. Such pregnancy complications include miscarriage, preeclampsia, fetal growth restriction, preterm birth and perinatal death [13,14,15,16,17]. It is estimated that 3% of pregnant women in high-income countries have chronic kidney disease (CKD), and its prevalence is rising due to an increased number of women entering into pregnancy at advanced age and/or with metabolic conditions, including obesity or diabetes [18,19]. Clinical management of pregnant women with kidney disease, either in the acute or chronic disease phase, is difficult due to the limited number of drugs approved for use during pregnancy. In recent decades, multiple efforts have been made to understand how maternal organs and body systems adapt to pregnancy [11,20,21,22,23,24,25]. However, the precise morphological, metabolic and molecular mechanisms occurring in the maternal kidney in response to pregnancy remain unknown. Addressing this key knowledge gap would be highly useful for the design and effectiveness of therapies to improve the immediate and life-long health of women with pregnancy and renal conditions.

Information on the structural, molecular and metabolic mechanisms mediating functional changes in the maternal kidneys in response to pregnancy is challenging in humans; therefore, research needs to be conducted in animal models. Although no animal model fully reproduces all aspects of human renal function in health and disease (e.g., differences in nephrogenesis [26] or some mouse strains are more resistant to glomerulosclerosis than others [27,28]), in recent decades, the use of mice for studies of kidney physiology has increased [28]. Moreover, pregnancy-related studies have mostly been conducted in mice due to their similarities with humans in terms of placentation [29], as well as their short gestational length compared to other common experimental animal models, such as sheep and pigs [30,31]. Hence, in this study, we investigated how the mouse renal system responds to pregnancy by analysing the structural and molecular changes in the maternal kidneys in normal pregnancy compared to the non-pregnant state. We also undertook mass spectrometry profiling of plasma and urine from non-pregnant and pregnant mice. The results obtained from these experiments allowed us to identify critical signalling pathways mediating morphological and functional changes in the maternal kidneys during pregnancy. We also generated a circulating and urine proteomic profile, identifying unique proteins in pregnancy that could be used to monitor and study potential mechanisms of renal adaptations during pregnancy.

## 2. Results

### 2.1. Pregnancy Increases Kidney Size

Experimental workflow and techniques employed in the study are shown in Figure 1A. Kidney weights increased by 15% during pregnancy compared to NP mice (Figure 1B). However, this renal expansion was proportional to the weight gain of the mice during pregnancy (kidney weight displayed as a proportion of hysterectomised body weight; Figure 1B). The increase in kidney mass during pregnancy was related to an expansion of the medulla and not the cortex (20% and 28% increase in medulla perimeter and volume, respectively, compared to NP mice; Figure 1C). Glomerular number and size in the cortex were similar between NP and pregnant mice (Figure 1D-F).

### 2.2. Pregnancy Modifies Cell Populations in the Kidney

Detailed histological examination of the kidneys revealed that in four out of five pregnant mice, there was an increase in the number of interstitial cell constituents, predominantly in the renal medulla and within the outer border, when compared to the NP mice (Figure 2A–D). In two of the pregnant animals analysed, the renal cortex also had higher numbers of interstitial cells adjacent to the cortical–medullary junction. Interstitial cells were morphologically compatible with sentinel lymphocytes, dendritic cells, endothelial cells, fibroblasts and histiocytes (Figure 2C,D) and were randomly located between collecting ducts, loops of Henle and distal tubules (i.e., they were not associated specifically with a certain structure or aggregated in specific locations). No pathological changes were noted in the renal corpuscles in any of the NP or pregnant mice. However, in three of the five pregnant mice, the diameter of glomerular tufts increased, and some tufts were split into several lobules with segmentally thickened basement membranes. This occurred in the absence of changes that would indicate impaired glomerular filtration capacity, such as protein casts (Figure 2E–H).

### 2.3. Pregnancy Increases Cell Proliferation in the Kidney

We next assessed whether structural changes in the kidney in response to pregnancy were related to changes in the proportion of proliferating and apoptotic cells using the Ki67-protein marker and TUNEL staining, respectively. In line with the expansion of the renal medulla, the percentage of proliferating cells in the outer and inner portion of the outer medulla was increased in the pregnant group by 63% and 61%, respectively, although no changes were observed in the inner medulla (Figure 3A). In addition, there was a 68% increase in cell proliferation in the glomeruli in pregnant compared to NP mice (Figure 3A). In contrast, the levels of cell apoptosis in the kidney were very low, and no differences were observed in any of the studied regions of the kidney between pregnant and NP mice (Figure 3B).

### 2.4. Pregnancy Is Associated with Changes in Key Cellular Signalling Pathways and in the mRNA Levels of Cyclin-Dependent Kinases (CDKs)

In order to identify the mechanisms through which the kidneys increase in size in pregnancy, we quantified the abundance of proteins in the PI3K-AKT, AMPK, PPAR and JNK-MAPKs signalling pathways, as they are involved in regulating growth, metabolism, proliferation and cell death [32,33,34,35,36,37,38]. The abundances of the insulin-signalling receptor (IR-β), PI3K-p110α and total AKT were unaltered, although PI3K-p85α and activated AKT (phosphorylated at S473) were significantly reduced by 79% and 68%, respectively, in the kidneys of pregnant mice compared to the NP group (Figure 4A). The abundances of PPAR-γ protein and energy sensor signalling protein, AMPK (total and phosphorylated levels) were significantly reduced in the pregnant kidney compared to the NP group (Figure 4B). Moreover, total JNK1 abundance and ERK1/2 activation (phosphorylated at T202/T204) were significantly reduced in the kidney in pregnancy by 75% and 56%, respectively, compared to NP mice (Figure 4C). However, total levels of ERK1/2 were increased by 42%, and total and phosphorylated levels of p38MAPK (phosphorylated at Thr180/Tyr182) increased by 34% and 200% in pregnant compared to NP females (Figure 4C).

The cell cycle is regulated, in part, by the cyclin-dependent kinases (CDKs) [39]. To assess whether they may contribute to the increase in renal size in pregnancy, we assessed the *Cdk* gene and related mRNA expression. We found that in response to pregnancy, mRNA levels of *Cdk1* (+28%)*, Cdc6* (+46%)*, Cdc42* (+13%) and *Ccne1* (+45%) were increased in the kidney compared to NP dams (Figure 4D). The expression of other cell proliferation-promoting genes, namely the transcriptional activator, *Foxm1,* and the tumour suppressor gene, *Rb1,* were increased by 43% and 21%, respectively (Figure 4D). Finally, the expression of mitochondrial biogenesis-related genes *Nrf1* and *Nrf2,* as well as the mitochondrial fusion gene *Mfn2*, were significantly elevated by ≈+22% when compared to NP mice (Figure 4E). No differences were found in the gene expression levels of *Cdk2, Cdk4*, *Tfam, Opa1, Mfn1, Drp1* or *Fis1* (Figure 4D,E).

### 2.5. Pregnancy Is Associated with Changes in Urine Proteomic Profile

To understand how pregnancy may affect the excretion function of the kidneys, we performed a qualitative LC-MS/MS analysis of urine samples collected from NP and pregnant mice. We detected a total of 237 proteins, of which 51 proteins were common between the two states (e.g., UROM, HEPC2 and CFAD; Appendix A). In contrast, 98 proteins were exclusively detected in the urine of NP mice, and 88 were present only in the urine of pregnant mice (Figure 5A and Appendix A). STRING analysis identified interactions between the signalling pathways altered in kidneys in response to pregnancy (PI3K-AKT, AMPK, PPAR and JNK-MAPKs; Figure 4A–C) and the proteins uniquely detected in the NP or the urine of pregnant mice (Figure 5A). STRING analysis also identified two well-defined clusters, namely keratins (e.g., KRT1, KRT10 or KRT4) and histones (e.g., H2AZ, H2AX or H2AV) among the proteins uniquely present in the urine of pregnant mice (Figure 5A). Overlapping the list of keratins and histones detected in mouse urine with a database of proteins identified in human urine [40] revealed 11 keratins in common (e.g., KRT1, KRT10 or KRT19), with none detected for urinary histones between mouse and human (Appendix A). Functional enrichment analysis revealed that proteins exclusively present in NP urine were implicated in tissue remodelling and cell migration related to collagen, extracellular matrix organization and the urokinase-type plasminogen signalling pathway (Figure 5B). On the other hand, the proteins uniquely identified in the urine of mice in the pregnant group were involved in keratin and intermediate filament organization, chromatin silencing and nucleosomal DNA binding (Figure 5C).

### 2.6. Pregnancy Is also Associated with Alterations in Plasma Proteomic Profile

In order to identify proteins that could be responsible for the changes observed in the pregnant kidney, we also analysed the plasma of NP and pregnant mice by LC-MS/MS [41]. This analysis identified 58 proteins with significantly varied abundance between NP and pregnant mice (Figure 6A and Appendix A). Informed by functional enrichment analysis, the proteins upregulated in pregnancy were implicated in processes such as complement activation and opsonin binding, whereas those downregulated in pregnancy were involved in negative regulation of lipoprotein activity and blood coagulation (Figure 6B,C). We then overlapped the 58 proteins that were altered (upregulated and downregulated) in the maternal circulation with the proteins present in only the NP urine (Figure 6D) and those present only in the pregnant urine (Figure 6E). By integrating these two datasets, we were able to identify proteins that were potentially altered in their reabsorption/excretion rate by the kidney in the pregnant mice relative to the NP state. This analysis revealed an overlap of five proteins in the NP urine (AMPB, CFB, IGFBP4, LYZ2 and THSB1) and four in the pregnant urine (GLYCAM1, ITIH4, SERPINA1B and SERPINA6) (Figure 6D,E). Using TissueEnrich [42] and UniProt gene ontology functional analysis, these nine overlapping proteins (5 for NP and 4 for pregnant urine) were identified to be mainly expressed by the liver, kidneys and lungs and to be involved in cell growth, immune system processes, cell adhesion and glucocorticoid metabolic processes (Table 1).

Finally, we assessed the contribution of the placenta as a source of the proteins detected in maternal plasma and urine during pregnancy. To this end, we overlapped the 58 proteins altered in the plasma in response to pregnancy with the recently published mouse placental secretome, comprising 314 proteins [41]. This analysis identified 16 common proteins between the two datasets (including IGFBP4, GPX3, VTN, C3 and C4B) (Figure 6F). We also overlapped the 88 unique urinary proteins in pregnant mice with the placental secretome and detected 6 proteins in common (including KRT1, PRSS2 and COL4A1) (Figure 6F).

## 3. Discussion

The current study shows that in mice, adaptations of maternal renal physiology during pregnancy are accompanied by changes in kidney structure, cellular proliferation and key genes and proteins involved in cellular homeostasis. The main structural changes were observed at the level of the renal medulla, with a predominant increase in interstitial cellular constituents due to cell proliferation. The renal medulla is essential for the regulation of urine concentration [43]. However, interstitial cells also play additional roles in protein synthesis, especially in the production of collagenous and non-collagenous extracellular proteins, including COL4A1 and COL5A2, which are both proteins found in pregnant urine and overlapped with the placental secretome. The medulla is also involved in the production of prostaglandins, such as PGE2 and PGF2α, which are known to be excreted in high concentrations in the urine of pregnant women [44,45].

In this study, we detected two well-defined clusters of proteins in the urine proteome: histones and keratins. Histones are important in gene regulation and DNA replication, and their presence in pregnant urine may indicate greater cellular proliferation and division occurring in the kidney [46]. On the other hand, keratins are cytoskeletal proteins involved in the regulation of cell shape, locomotion and stress [47,48]. KRT19 was exclusively detected in pregnant urine, representing a keratin that is upregulated upon tubular epithelial cell injury [47]. Detection of keratins in the urine may reflect the remodelling processes occurring in the kidney, as well as the mild glomerular splitting and segmentally thickened basement membranes reported in pregnant mice. The dominant histone and keratin urinary cluster is also consistent with the elevated activity of the p38MAPK signalling pathway in the pregnant kidneys. p38MAPK signalling regulates cell growth, stress and inflammation and is also involved post-transcriptional control of keratins [49,50,51]. Moreover, p38MAPK regulates the cell cycle by functioning as a CDK-like kinase [52], which may correlate with the increased mRNA levels of *Cdk1*, *Cdc6* or *Ccne1* detected in pregnant kidneys. Activation of this pathway may be also related to the increased number of immune interstitial cells (sentinel lymphocytes, dendritic cells and histiocytes) found in pregnant kidneys. The abundance of proteins involved in immune and stress responses (complement activation and acute-phase response), including SERPINA6, GLYCAM1 and ITIH4, were found to be increased in the circulation of pregnant compared to non-pregnant mice. ITIH4 is positively regulated by interleukin (IL)-6, a potent inflammatory cytokine that stimulates p38MAPK phosphorylation [53,54]. IL-6 concentrations are increased naturally in the circulation of mice in the leadup to term [55] and likely relate to changes in renal immune cell populations, as well as activation of the p38MAPK pathway during pregnancy. Our results, which showed partial overlap of proteins, including keratins detected in human urine (incomplete overlap probably due to species differences and variation in the period of pregnancy studied), could have important clinical relevance, as alterations in inflammatory processes play a key role in the development of preeclampsia, a pregnancy complication that can cause kidney injury/disease in the mother [56,57].

In addition to the changes observed in the p38MAPK signalling pathway, we observed alterations in other signalling pathways. We observed reduced activation of cell growth and metabolic pathways PI3K-AKT, ERK, AMPK and PPAR in the maternal kidneys in response to pregnancy. These adaptive changes may be protective, as growing evidence suggests that excessive activation of PI3K-AKT signalling can cause renal fibrosis and kidney dysfunction [34]. On the other hand, AMPK signalling inhibits cell growth and translation; therefore, reduced AMPK activation would favour renal growth [58]. AMPK controls lipid homeostasis and mitochondrial dynamics [59,60], and work is required to assess the relationships between reduced renal AMPK activation and the down-regulation of proteins implicated in the lipoprotein activity that we detected in the blood of pregnant mice. Activity of AMPK is promoted by PPAR-γ, which was also reduced in the maternal kidneys in response to pregnancy. PPAR-γ is expressed in the medullary collecting duct, podocytes, mesangial cells and vascular endothelial cells [58]. Studies using PPAR-γ agonists, such as pioglitazone or ciglitazone, have shown that activation of PPAR-γ signalling inhibits mesangial cell proliferation and ERK phosphorylation [61,62]. Thus, reduced levels of PPAR-γ may additionally have attenuated ERK activation in the kidneys of pregnant mice. Therefore, these data suggest that during pregnancy, classical signalling cascades/pathways change through potential synergistic and antagonistic interactions with the aim of ensuring adequate renal function and preservation of maternal and foetal well-being. However, delineating the contribution of each signalling pathway in the regulation of maternal kidney adaptations in pregnancy will be challenging and would require a combination of mouse genetic lines and in vitro experiments.

We found that adaptations in maternal renal physiology are related to specific types of proteins in the blood and urine during pregnancy, which may be suitable for the development of diagnostic biomarkers of pregnancy well-being and kidney function more generally. For instance, IGFBP4 was present in NP urine but undetectable in urine of pregnant mice. Our data are consistent with studies of IGFBP4 in human non-pregnant and pregnant urine [40] and suggest an increase in the reabsorption of the protein by the kidney during pregnancy. IGFBP4 is secreted by the placenta, a main source in pregnancy [41], but it is also produced by the liver and the mesangial cells of the kidneys [63]. Overexpression of IGFBP4 in primary renal cancer cells promotes cell proliferation, invasion and migration [64]. These observations suggest that IGFBP4 could be one of the key regulators of renal growth during pregnancy. Future studies could evaluate the contribution of IGFBP4 in mediating the pregnancy-induced changes in renal physiology by using IGFBP4-null mice [65]. Finally, we observed that the placenta is a source of such proteins [41] and likely interacts at the level of the kidney during pregnancy. Elucidating the nature of interorgan crosstalk between the placenta and the kidney will be valuable for identifying therapies to prevent and treat pregnancy disorders, such as preeclampsia. Indeed, preeclampsia is the leading cause of nephrotic syndrome during pregnancy [66], and defects in the placenta are a primary cause of the ‘great obstetrical’ complications, which include preeclampsia [67]. Our current work paves the way for future research lines, as we have described physiological adaptations in the mouse kidney without placental disease. Future work should therefore utilize rodent models of placental malfunction (e.g., induced via genetic, dietary or surgical methods) [68,69,70,71,72] to improve our knowledge on the pathophysiology of renal and pregnancy diseases.

In summary, we have demonstrated that in response to pregnancy, the maternal kidney adapts structurally through changes in the activity of key cellular signalling pathways. Our work also identifies novel proteins secreted by the placenta and by other maternal organs contributing to the adaptations in maternal body systems required for adequate pregnancy. We highlight p38MAPK and IGFBP4 as potential key signalling and functional proteins mediating these adaptations. Future work should evaluate the function of these enzymes/pathways under different conditions, such as in a pregnancy time-course study, in the post-partum period or in animal models of complicated gestations. This work will ultimately help to identify potential therapeutic options for women with kidney diseases and/or pregnancy complications.

## 4. Materials and Methods

### 4.1. Animal Work

All animal work was performed under the U.K. Animals (Scientific Procedures) Act 1986 following ethical approval by the University of Cambridge. No sample size calculation was conducted prior to undertaking this study. However, the number of animals employed for the study (N value) was chosen based on accepted practices and based on related published work in the field [41]. Experiments were conducted at the University of Cambridge Animal Facility under a 12/12 dark/light system. Eight- to ten-week-old C57BL6/J female mice were fed ad libitum with a standard chow diet (RM3; Special Dietary Services). Females were divided in two groups: non-pregnant virgin females (NP) and pregnant dams. Females in the pregnant group were time-mated with C57BL6/J males, and the day a copulatory plug was detected was designated as gestational day (GD) 1. At 09:00 h, urine was collected from NP and pregnant mice (on GD16) by gently holding/scuffing the mouse and using a 50 mL falcon tube. Tubes were directly immersed in liquid nitrogen and stored at −80 °C until analysis. Mice were killed by cervical dislocation; the uterus was removed; and the kidneys were harvested, rinsed in PBS and then collected for histological (right kidney) and molecular analysis (left kidney). For pregnant mice, GD16 was specifically chosen, as it corresponds to 80% of total mouse pregnancies (term on GD20), when the foetus enters its most rapid growth phase. Moreover, this state is approximately equivalent to the start of the last trimester of human pregnancy. 

### 4.2. Histology

Kidneys used for histological analysis were processed with two different methods according to the parameter assessed. In order to determine the perimeter, area and volume of the renal cortex and medulla, kidneys were cut along their longitudinal axis, using the renal artery and vein insertions as reference points. Then, kidneys were embedded in OCT cryo-embedding matrix (CellPath), as previously described in [73], and cut to a thickness of 12 μm with a cryostat. This method was preferred over paraffin sections to avoid dehydration of the samples. After cutting the sections, samples were washed in PBS, fixed in 4% paraformaldehyde, stained with hematoxylin-eosin (H–E) using standard protocols and scanned with a nanozoomer scanner (Hamamatsu). Analysis was performed using the freehand region tool of NDPI software. The volume of the cortex or medulla was corrected to total renal size and multiplied by the weight of the kidney.

For histopathological analysis, immunohistochemistry and glomerular size assessment, we worked with paraffin-embedded sections. Briefly, kidneys were fixed overnight in 4% paraformaldehyde, dehydrated through ethanol incubations and paraffin-embedded according to standard protocols for mouse tissues [74]. Paraffin blocks were sectioned with 3 μm thickness and stained in periodic acid–Schiff (*PAS*) and H-E. Sections were imaged with a nanozoomer scanner (Hamamatsu), and glomerular number and size were determined in ≥50 randomly selected cortical glomeruli. Histopathological analysis was performed by a trained histopathologist.

#### Immunohistochemical Analysis of Ki67 and TUNEL Staining

Paraffin-embedded sections were used for Ki67 immunostaining (ab264429, Abcam, Cambridge, UK) to indicate cell proliferation. Briefly, sections were dewaxed in xylene and rehydrated by reducing concentrations of ethanol to distilled water. Antigen retrieval was performed in a microwave with sodium citrate buffer (pH 6.0). Endogenous peroxidase activity was quenched by incubating the slides with 3% hydrogen peroxide diluted in methanol, and washes were performed with tris-buffered saline supplemented with tween-20 (TBS-T). Blocking was performed with 10% goat serum, and antibody was applied overnight at 4 °C (1:500 in 5% goat serum diluted in TBS-T). The next day, slides were washed in TBS-T and incubated with goat anti-rabbit secondary antibody (1:1000, ab6720, Abcam) and streptavidin-horseradish peroxidase (1:500, S000-03, Rockland, PA, USA). Sections were stained with 3,3′-diaminobenzidine (ab64238, Abcam) and counterstained with nuclear fast red (H-3403-500, Vector). The level of cell apoptosis was determined using an HRP-DAB TUNEL assay kit (ab206386, Abcam) according to the manufacturer’s instructions. Ki67- and TUNEL-positive cells were determined in 100 randomly selected glomeruli and expressed as a ratio of the glomerular size.

### 4.3. Western Blot Analysis

Protein was extracted using RIPA buffer (R0278, Sigma-Aldrich, St. Louis, MO, USA). Extracted protein was diluted to a constant concentration (3 μg/μL) in SDS loading buffer and denatured at 90 °C for 5 min before separation by acrylamide gel electrophoresis. Proteins were then transferred from acrylamide gels onto nitrocellulose membranes (1620115, Bio-rad, Hercules, CA, USA) and blocked either with 5% foetal bovine serum (Sigma-Aldrich) or semi-skimmed milk (Marvel, New York, NY, USA) for 1 h at room temperature. Membranes were incubated overnight at 4 °C with specific primary antibodies (Appendix A). The next day, membranes were washed with TBS-T and incubated with secondary antibodies (1:10,000 NA934-1ML or NA931-1ML, Amersham, St. Louis, MO, USA). Blots were washed in TBS-T and subsequently exposed to ECL substrate (SuperSignal West Femto, Thermofisher, Waltham, MA, USA). Images of the blots were taken with an Invitrogen iBright imaging system. Band intensity was calculated with ImageJ, and protein abundance was normalized to total protein content using Ponceau S staining [75].

### 4.4. RNA Extraction and qPCR

RNA was extracted with an RNeasy Plus mini kit (Qiagen, Hilden, UK) according to the manufacturer’s instructions. A total of 2 μg of RNA per sample was reverse-transcribed using a high-capacity cDNA reverse transcription kit (Applied Biosystems, Foster City, CA, USA) according to the manufacturer’s instructions and as previously described for mouse tissue [76]. The expression of genes of interest was determined by qPCR (StepOne real-time PCR system, ThermoFisher) in three dilutions of each cDNA sample (1:10, 1:20 and 1:100) using SYBR Green master mix (Applied Biosystems, Cheshire, UK) and primers described in Appendix A. Relative expression was calculated using the 2^−ΔΔCt^ method [77], and genes of interest were normalized to the geometric mean expression of housekeeping genes (*Hprt*, *Ywhaz, Polr2a* and *Ubc*), which were not altered by pregnancy state.

### 4.5. LC-MS/MS Analysis of Urine and Plasma Samples

Two types of LC-MS/MS were performed. For urine analysis, we conducted a qualitative LC-MS/MS based on the presence or absence of peptides. For maternal plasma, we used LC-MS/MS to perform a relative quantitation-based analysis of plasma proteins, whereby peptide peak areas from target proteins were normalized to the bovine insulin internal standard. Methods related to the plasma LC-MS/MS were previously described in detail [41]. For urine analysis, 800 µL of water was added to 200 µL of urine. The samples were centrifuged to remove any particulate and then loaded directly onto a solid-phase extraction plate and extracted and tryptically digested as described in [41].

Protein identification was performed using PEAKS 8.5 (BSI, Waterloo, ON, Canada) using the UniProt database, and proteins identified in the LC-MS/MS analyses were converted from UniProtKB AC/ID into gene names. Analyses of proteins were performed with STRING (https://string-db.org/) and full STRING networks were built using gene names and a medium confidence interaction score with the ‘hide disconnected nodes’ setting enabled. Colours were modified with Adobe Illustrator to facilitate visualization of networks. Network node analysis was performed using functional enrichment analysis based on the biological process, molecular function and cellular components of the nodes detected. Gene enrichment analyses were conducted using TissueEnrich (https://tissueenrich.gdcb.iastate.edu/).

### 4.6. Statistical Analysis

GraphPad Prism software (version 9) was used to determine statistical differences between the NP and pregnant groups. Normality of the data was assessed with Shapiro–Wilk test. Differences between groups were analysed by parametric unpaired Student’s *t*-test (normally distributed) or non-parametric Mann–Whitney U test (not normally distributed). Statistical analysis for the plasma proteomics data was performed with Perseus software [78], and significant proteins were identified with a permutation-based correction controlled by an FDR threshold of 0.05. All data are reported as mean ± SEM, and the number of animals and specific statistical analysis for each variable is reported in figure legends.

## Figures and Tables

**Figure 1 ijms-23-06287-f001:**
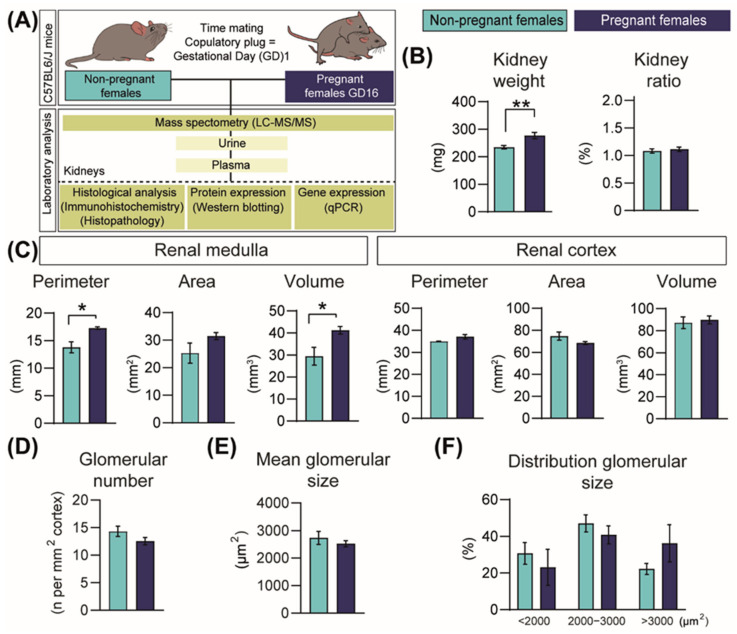
Structural changes in the mouse kidney in response to pregnancy in mice. (**A**) Experimental design. (**B**) Kidney weight and kidney size normalized to hysterectomised weight (*n* = 10/group). (**C**) Renal medulla and cortex sizes (*n* = 4–5/group). (**D**,**F**) Glomerular number and glomerular size in the renal cortex (*n* = 5/group). All data are presented as mean ± SEM and were analysed by unpaired Student *t*-test or Mann–Whitney U Test. Asterisks represent significant difference between non-pregnant and pregnant mice (* *p* < 0.05; ** *p* < 0.01).

**Figure 2 ijms-23-06287-f002:**
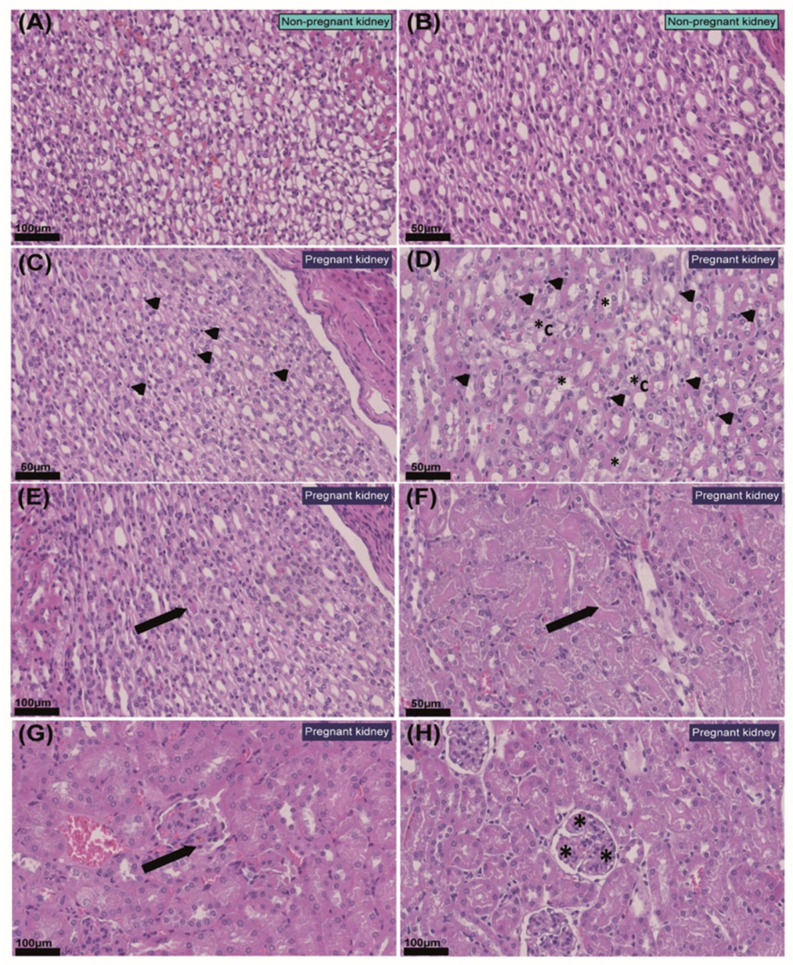
**Histopathological changes in the kidney in response to pregnancy in mice.** (**A**,**B**) Representative images of non-pregnant renal medulla. (**C**) Renal medulla of pregnant mouse with mildly increased numbers of interstitial cellular constituents (arrow heads). (**D**) Renal medulla of pregnant mouse with moderately increased numbers of cells within the interstitium (arrow heads), distal tubule epithelial degeneration (*) and collecting duct epithelial degeneration (*C). (**E**) Protein casts within proximal tubules with mild degeneration in the renal cortex of a pregnant mouse (arrow). (**F**) Protein casts within distal tubules in renal medulla of a pregnant mouse (arrow). (**G**) Segmental thickening of the glomerular basal membrane in pregnant kidney (arrow). (**H**) Mild glomerular splitting in pregnant kidney (*). Values of scale bars are shown in the images.

**Figure 3 ijms-23-06287-f003:**
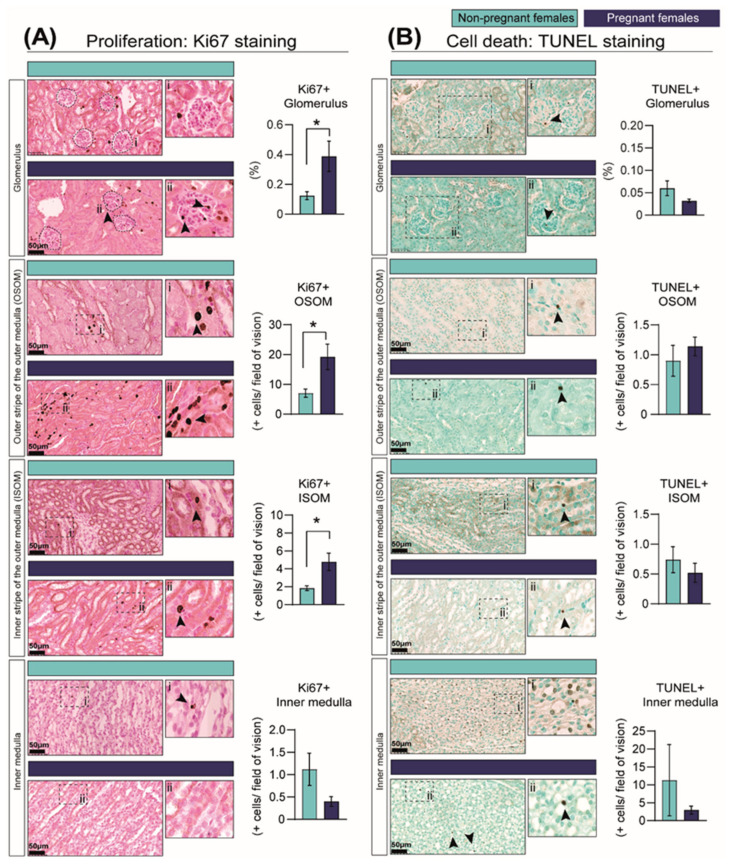
**Percentage of cells showing cell-cycle activation as informed by Ki67 immunostaining (A) and cell death as identified by TUNEL assay (B) in the kidney in response to pregnancy in mice.** Representative image of stained kidney from non-pregnant and pregnant female mice. Data are presented as mean ± SEM (*n* = 5/group). Asterisks represent significant differences between non-pregnant and pregnant mice as determined by Student’s *t*-Test (* *p* < 0.05). Images with the labels i and ii depict high magnification of the selected area. Arrow heads indicate positive DAB staining. Scale bar is 50 µm.

**Figure 4 ijms-23-06287-f004:**
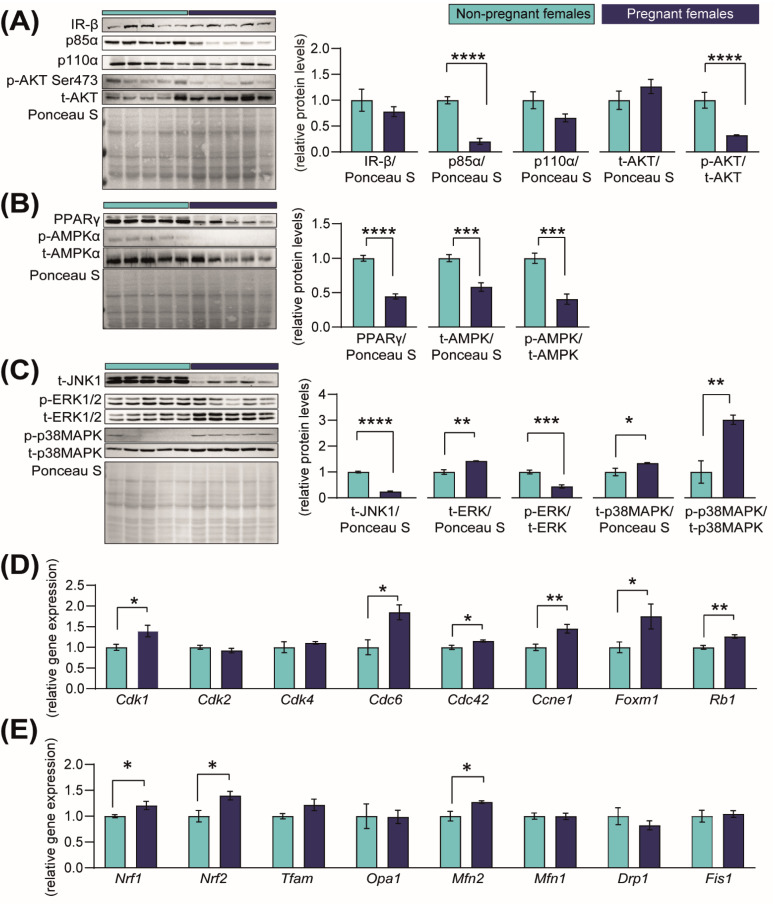
**Abundance of proteins in growth-signalling pathways and expression of cell-cycle regulatory genes in the kidney in response to pregnancy in mice.** (**A**–**C**) Immunoblots and quantification of PI3K-AKT, PPAR-AMPK and MAPK-JNK signalling proteins in the kidneys of non-pregnant and pregnant female mice (*n* = 5/group) (**D**,**E**). Expression of genes related to cell-cycle progression and mitochondrial-related genes in kidneys of non-pregnant and pregnant female mice (*n* = 4–5/group). Data are presented as mean ± SEM and were analysed by unpaired Student’s *t*-test. Asterisks represent significant differences between non-pregnant and pregnant mice (* *p* < 0.05; ** *p* < 0.01; *** *p* < 0.001; **** *p* < 0.0001).

**Figure 5 ijms-23-06287-f005:**
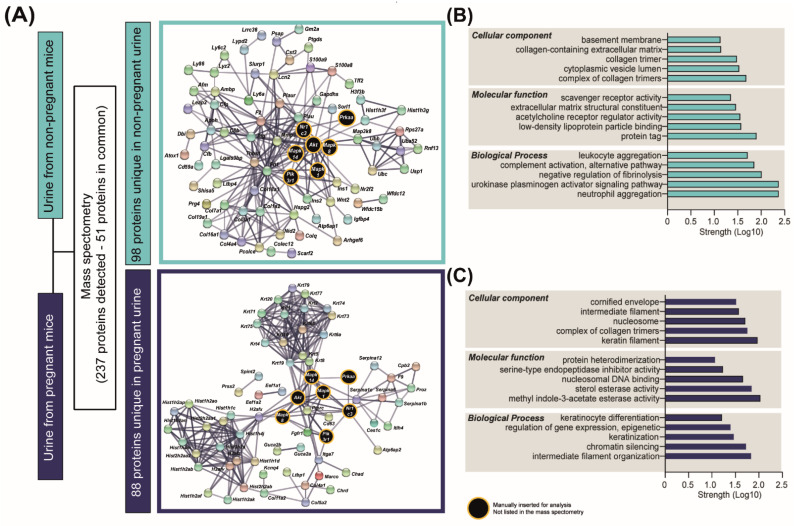
**Urinary proteins detected in non-pregnant and pregnant mice.** (**A**) Results of the analysis of proteins detected by mass spectrometry in urine samples (*n* = 3/group). (**B**,**C**) Top-scoring cellular, molecular and biological processes for the 98 and 88 proteins uniquely detected in non-pregnant and pregnant urine, respectively. Network representation and pathway analysis were performed with STRING.

**Figure 6 ijms-23-06287-f006:**
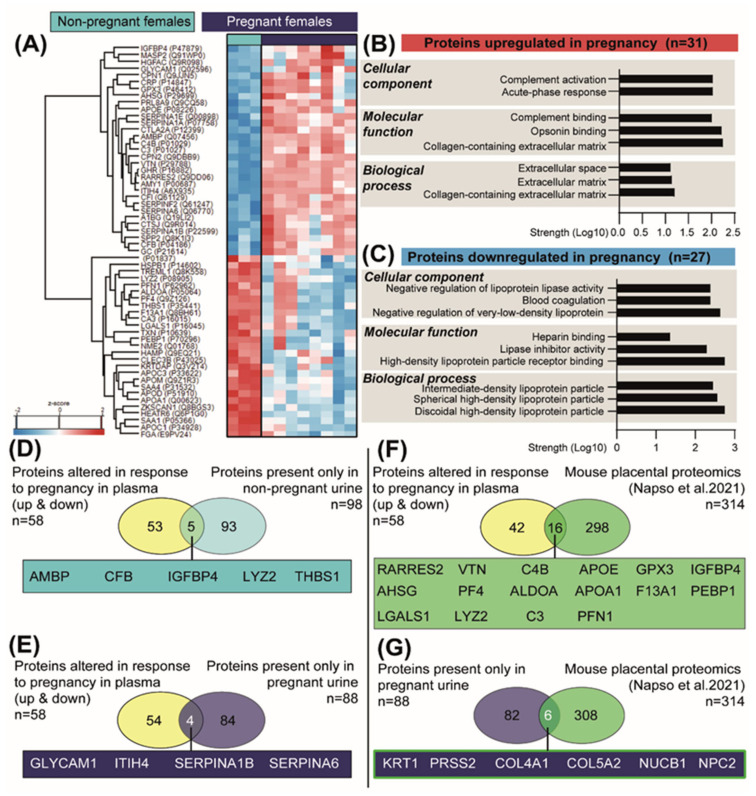
**Changes in the abundance of plasma proteins in response to pregnancy in mice.** (**A**) Heat map of differentially expressed plasma proteins between the pregnant and non-pregnant mice, as detected by mass spectrometry (*n* = 3 non-pregnant and *n* = 8 pregnant). (**B**,**C**) Top-scoring cellular, molecular and biological processes for the 31 and 27 proteins upregulated and downregulated in pregnant mice, respectively. Analysis performed with STRING. (**D**) Venn diagram showing the overlay of proteins altered in plasma in response to pregnancy (upregulated and downregulated) with the proteins exclusively detected in the non-pregnant urine. (**E**) Venn diagram showing the overlay of altered proteins in plasma in response to pregnancy (upregulated and downregulated) with the proteins exclusively detected in the pregnant urine. (**F**) Venn diagram showing the overlay of altered proteins in plasma in response to pregnancy (upregulated and downregulated) with proteins previously reported to be secreted by the mouse placenta [41]. (**G**) Venn diagram showing the overlay of proteins detected in the pregnant urine with proteins previously reported to be secreted by the mouse placenta [41].

**Table 1 ijms-23-06287-t001:** **Tissue enrichment and function of the nine proteins altered in both urine and plasma in response to pregnancy.** Analysis was performed with TissueEnrich (https://tissueenrich.gdcb.iastate.edu/) and UniProt (https://www.uniprot.org/).

Protein Name	Direction of the Change in Pregnancy	Presence in the Urine	Top Three Organs—Enrichment (Expression Levels Based on FPKM)	GO Function Biological Process
AMBP	↑	NP	Liver (465.23) Heart (3.41) Bone marrow (1.87)	Receptor-mediated endocytosis and cell adhesion
CFB	↑	NP	Liver (110.86) Kidney (38.95) Intestine (18.33)	Immune system process and proteolysis
IGFBP4	↑	NP	Liver (231.39) Kidney (151.64) Spleen (33.41)	Regulation of cell growth and inflammatory response
LYZ2	↓	NP	Lung (1308.75) Intestine (337.09) Spleen (283.58)	Inflammatory response and metabolic process
THBS1	↓	NP	Bone Marrow (13.92) Lung (5.57) Cortex (4.49)	Activation of MAPK activity and response to hypoxia
GLYCAM1	↑	Pregnant	Spleen (0.28) Kidney (0.20) Intestine (0.18)	Regulation of immune response
ITIH4	↑	Pregnant	Liver (114.27) Lung (3.43) Heart (0.33)	Platelet degranulation and acute-phase response
SERPINA1B	↑	Pregnant	Liver (844.81) Heart (6.34) Kidney (5.08)	Platelet degranulation and ER to Golgi vesicle-mediated transport
SERPINA6	↑	Pregnant	Liver (332.31) Heart (0.28) Bone marrow (0.20)	Glucocorticoid biosynthetic and metabolic processes

## Data Availability

All relevant data are presented within the paper and available upon reasonable request.

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
