# Peer review of "Identification of Structural and Molecular Signatures Mediating Adaptive Changes in the Mouse Kidney in Response to Pregnancy"

_ijms, 2022, doi:10.3390/ijms23116287_

Round 1

Reviewer 1 Report

The authors investigated how the renal system responds to pregnancy by analysing structural and functional changes in kidney between non-pregnant and pregnant mice. Further, the authors identified signalling pathways, which may be involved in maternal kidney adaptation. The findings in this study may contribute to develop diagnostic biomarkers for maternal and foetal health in human pregnancy ultimately. The experiments in this study are designed well and the data support reasonably the conclusion. The following is a minor comment, which I hope to improve this manuscript further.

Comment:

As far as I know, detecting Karatin and Histones in urine during pregnancy does not happen in human. Since this study aims to understand human pregnancy using mice, it would be good to discuss a bit the difference between mice and human in urine during pregnancy.

Author Response

Authors: We appreciate the reviewers’ feedback on our manuscript. This has helped us to improve our manuscript. Our point-by-point response to each reviewer’s comment is detailed below and changes in the manuscript are highlighted in red.

Question from reviewer: As far as I know, detecting Keratin and Histones in urine during pregnancy does not happen in human. Since this study aims to understand human pregnancy using mice, it would be good to discuss a bit the difference between mice and human in urine during pregnancy.

Authors: Work by Zheng et al., 2013, has shown that histones and keratins are present in the urine of women during pregnancy (DOI: 10.1186/1471-2164-14-777; see supplementary Table 3 – last spreadsheet). They identified 32 keratins in human urine (7 of which were exclusively detected in pregnancy). They also identified 37 histones in human urine, of which 4 were exclusively detected in pregnant urine. We have now overlapped all our list of keratins and histones in mouse urine with those detected by Zheng et al in humans and found several keratins (but not histones) in common. We have inserted these new findings in the results section and the revised discussion of the manuscript.

“STRING analysis also identified two well-defined clusters, namely keratins (eg. KRT1, KRT10, or KRT4) and histones (eg. H2AZ, H2AX or H2AV) for the proteins uniquely present in the urine of pregnant mice (Figure 5A). Overlapping the list of keratins and histones detected in mouse urine with a database of proteins identified in human urine [46] revealed 11 keratins in common (eg. KRT1, KRT10 or KRT19), with none detected for urinary histones between mouse and human (Supplementary Table 3).

Our results, which showed partial overlap with proteins including keratins detected in human urine (incomplete overlap probably due to species differences and variation in the period of pregnancy studied), could have important clinical relevance, as alterations in inflammatory processes plays a key role in the development of preeclampsia, a pregnancy complication that can cause kidney injury/disease in the mother [61,62].”

Reviewer 2 Report

Lopez-Tello et al. present a very interesting manuscript, well written and structured. The authors carry out a solid experimental design that makes the results also solid. The results are adequately described and explained sequentially, all of which makes the results support the conclusions. Suggest minor points to authors:

-In the first place, the title should be modified and be more specific, it is too generic a title and it does not have enough force.

-In keywords, authors should include a more specific word.

-Review the phrase on lines 65-75. At this point the authors are a bit confused, I suggest including some reference.

-Review point 2.2., include specific references.

-Review point 2.4, include specific references.

-Figure 2 and 3 should be better explained in the manuscript. Histological images should be better described in the text.

-The authors must justify the sample size.

-The discussion should have a more translational character. The authors should include more specifically aspects related to placental pathology.

--The authors must include a graphic abstract.

Author Response

We appreciate the reviewers’ feedback on our manuscript. This has helped us to improve our manuscript. Our point-by-point response to each reviewer’s comment is detailed below and changes in the manuscript are highlighted in red.

-In the first place, the title should be modified and be more specific, it is too generic a title and it does not have enough force.

We agree with the reviewer and we have modified the title of the manuscript. The title reads now: “Identification of structural and molecular signatures mediating adaptive changes in the mouse kidney in response to pregnancy

-In keywords, authors should include a more specific word.

We agree with the reviewer, we have inserted new keywords (highlighted in red in the new version of the manuscript)

Keywords: Pregnancy, Kidney, Physiology, Signalling, Renal Medulla, Mouse Urine, Mass Spectrometry, Keratins, Proliferation, Interstitial Cells

-Review the phrase on lines 65-75. At this point the authors are a bit confused, I suggest including some reference.

We have now included the following text and citations (highlighted in red):

“Information on the structural, molecular and metabolic mechanisms mediating functional changes in the maternal kidneys in response to pregnancy is challenging in humans and therefore, research needs to be conducted in animal models. Although no animal model fully reproduces all aspects of human renal function in health and disease (eg. differences in nephrogenesis [26]  or some mouse strains are more resistant to glomerulosclerosis than others [27,28]), in the last decades the use of mice for studies of kidney physiology has increased [28]. Moreover, pregnancy related studies are mostly conducted in mice due to their similarities in placentation with humans [29], and their short gestational length compared to other common experimental animal models, like sheep and pig [30,31].

-Review point 2.2., include specific references.

 We have included the following references highlighted in red:

 “Kidneys used for histological analysis were processed in two different ways according to the parameter assessed. In order to determine the perimeter, area and volume of the renal cortex and medulla, kidneys were cut along their longitudinal axis, using the renal artery and vein insertions as reference points. Then, kidneys were embedded in OCT cryo-embedding matrix (CellPath) as previously described by [33], and cut at 12μm thick with a cryostat.”

 Briefly, kidneys were fixed overnight in 4% paraformaldehyde, dehydrated through ethanol incubations and paraffin embedded according to standard protocols for mouse tissues [34].”

 -Review point 2.4, include specific references.

RNA was extracted with the RNeasy Plus Mini Kit (Qiagen, Hilden, UK) according to manufacturer’s instructions. A total of 2μg of RNA per sample was reverse transcribed using the High-Capacity cDNA Reverse Transcription Kit (Applied Biosystems, Foster City, USA) according to the manufacturer’s instructions and as previously described for mouse tissue [36]. The expression of genes of interest was determined in three dilutions of each cDNA sample (1:10, 1:20 and 1:100) using SYBR Green master-mix (Applied Biosystems, UK) and primers described in Supplementary Table 2 by qPCR (StepOne real-time PCR system, ThermoFisher). Relative expression was calculated using the 2-ΔΔCt method [37] and genes of interest were normalized to the geometric mean expression of housekeeping genes (Hprt, Ywhaz, Polr2a and Ubc), which were not altered by pregnancy state.”

 -Figure 2 and 3 should be better explained in the manuscript. Histological images should be better described in the text.

As suggested, we have re-written and expanded the section describing histopathological changes in the kidney.

The text reads now:

 Pregnancy modifies cell populations in the kidney

Detailed histological examination of the kidneys revealed that in 4 out of 5 pregnant mice there was an increase in the number of interstitial cell constituents, predominantly in the renal medulla and within the outer border when compared to the NP mice (Figure 2A-D). In 2 of the pregnant animals analysed, the renal cortex also had higher numbers of interstitial cells, adjacent to the cortical-medullary junction. Interstitial cells were morphologically compatible with sentinel lymphocytes, dendritic cells, endothelial cells, fibroblasts and histiocytes (Figure 2C-D), and were randomly located between collecting ducts, loops of Henle and distal tubules (i.e. they were not associated specifically to a certain structure or aggregated in specific locations). No pathological changes were noted in the renal corpuscles in any of the NP and pregnant mice. However, in 3 of the 5 pregnant mice, glomerular tufts were increased in diameter and some tufts were split in several lobules with segmentally thickened basement membranes. This was in the absence of changes that would indicate impaired glomerular filtration capacity, such as protein casts (Figure 2E-H).

 We have also modified the figure legend for figure 3, now it reads:

Figure 3. Percentage of cells showing cell cycle activation as informed by Ki67 immunostaining and cell death as identified by TUNEL assay in the kidney in response to pregnancy in mice.

 -The authors must justify the sample size.

In response to this comment, we have added the following section in the materials and methods.

“No sample size calculation was conducted prior to undertaking this study. However, the number of animals employed for the study (N values) were chosen based on accepted practices and based on related published work in the field [32].”

 -The discussion should have a more translational character. The authors should include more specifically aspects related to placental pathology.

We have incorporated one paragraph into the discussion. We agree with the reviewer that placental pathology is one cause of kidney disease (especially in preeclampsia). However, in our model, we do not have any pathological condition. Therefore, the new section of the manuscript is focused on future work. We hope the reviewer agrees with our new text.

“Finally, we have observed that the placenta is a source of such proteins [32] and likely interacts at the level of the kidney during pregnancy. Elucidating the nature of inter-organ crosstalk between the placenta and the kidney will be valuable for identifying therapies to prevent and treat pregnancy disorders, such as preeclampsia. Indeed, preeclampsia is the leading cause of nephrotic syndrome during pregnancy [72] and defects in the placenta are a primary cause of the ‘great obstetrical’ complications, which includes preeclampsia  [73]. Our current work paves the way for future research lines, as we have described physiological adaptations in the mouse kidney without placental disease. Future work should therefore utilize rodent models of placental malfunction (e.g. induced via genetic, dietary or surgical methods) [74–78] to improve our knowledge on the pathophysiology of renal and pregnancy diseases.”

 -The authors must include a graphic abstract.

We have added a graphical abstract as suggested (please see the PDF).
